# Effectiveness of intralesional sodium stibogluconate for the treatment of localized cutaneous leishmaniasis at Boru Meda general hospital, Amhara, Ethiopia: Pragmatic trial

**Feleke Tilahun Zewdu**[1]*, **Asressie Molla Tessema**[2], **Aregash Abebayehu Zerga**[2], **Saskia van Henten**[3], **Saba Maria Lambert**[4]¤

**1** Boru Meda general hospital, Dermatology clinic, Dessie, Ethiopia, **2** Wollo University, Department of Public health, Dessie, Ethiopia, **3** Institute of Tropical Medicine, Antwerp, Belgium, **4** London School of Hygiene & Tropical Medicine, London

¤ Current address: ALERT, Dermatology Department, Addis Ababa, Ethiopia
* momflk@gmail.com

## Abstract

### Background

Cutaneous leishmaniasis (CL) is generally caused by *Leishmania aethiopica* in Ethiopia, and is relatively hard to treat. Sodium stibogluconate (SSG) is the only routinely and widely available antileishmanial treatment, and can be used systemically for severe lesions and locally for smaller lesions. There is limited data on the effectiveness of intralesional (IL) SSG for localized CL in Ethiopia and therefore good data is necessary to improve our understanding of the effectiveness of the treatment.

### Methodology/Principal findings

A pragmatic (before and after Quazi experimental) study was done to assess the effectiveness of intralesional SSG among localized CL patients at Boru Meda general hospital, Northeast Ethiopia. Patients who were assigned to intralesional SSG by the treating physician were eligible for this study. Study subjects were recruited between January and August 2021. Infiltration of intralesional SSG was given weekly to a maximum of six doses. However, when a patient's lesions were already cured before getting 6 doses, treatment was not coninued, and patient were only asked to come for lesion assessment. Skin slit smears (SSS) were taken each week until they became negative. Outcomes were assessed at day 90, with patients who had 100% reepithelization (for ulcerative lesions) and/or flattening (for indurated lesions) defined as cured. Multi-level logistic regression was done to assess factors associated with cure.

A total of 83 patients were enrolled, and final outcomes were available for 72 (86.75%). From these 72, 43 (59.7%, 95% confidence interval 0.44–0.69) were cured at day 90. Adverse effects were common with 69/72 patients (95.8%) reporting injection site pain.

**Data Availability Statement:** All relevant data are within the manuscript and its Supporting Information files.

**Funding:** The author(s) received no specific funding for this work.

**Competing interests:** The authors have declared that no competing interests exist.

Factors associated with cure were age (OR 1.07 95% CI: 1.07–1.27), being male (OR 1.79, 95% CI: 1.10–2.25), size of the lesion (OR 0.79, 95% CI: 0.078–0.94) and skin slit smear (SSS) result +1 grading (OR 1.53, 95% CI: 1.24–1.73) and +2 grading (OR 1.51, 95% CI: 1.41–3.89) compared to the SSS grade +6.

## Conclusion

Our findings revealed that intralesional sodium stibogluconate resulted in a cure rate of around 60%, with almost all patients experiencing injection site pain. This emphasizes the need for local treatment options which are more patient-friendly and have better cure rates.

### Author summary

Cutaneous leishmaniasis (CL) is a neglected tropical disease that is quite common in Ethiopia. This is caused by *Leishmania aethiopica*, which results in severe lesions that do not seem to respond well to treatment. In Ethiopia, sodium stibogluconate is the most widely used treatment for leishmaniasis, but data on the effectiveness of intralesional sodium stibogluconate for localized CL is scarce. We recorded treatment outcomes and adverse events of patients with localized CL lesions who were treated with intralesional SSG up to six times with one injection administered every week. Patients were followed for 90 days, after which the cure rate was assessed. Among 83 patients enrolled, 72 were seen for outcome assessment. Less than 60% of patients were cured, despite having received at least 5 doses of treatment. Almost all patients reported injection site pain as a side-effect. The low cure rate and a high proportion of patients reporting pain indicate that intralesional sodium stibogluconate is not ideal as a treatment regimen. Cryotherapy or thermo therapy could be a better alternative, but a comparative clinical trial is needed to provide concrete evidence as to which treatment should be recommended.

## Introduction

Cutaneous leishmaniasis (CL) is a parasitic disease caused by bites of infected sand flies. The annual incidence is estimated between 0.7 and 1.2 million people in over 90 countries worldwide [1]. CL in Ethiopia is estimated to affect 20,000 and 40,000 persons per year [1,2] and more than 28 million persons are at risk of infection [3–4]. *Leishmania aethiopica* is the causative species in Ethiopia, and lesions are often severe, diffused, erythematous, and difficult to treat compared to lesions caused by other species [5].

CL in Ethiopia has a wide range of clinical presentations [4,6], with lesions presenting as an erythematous patch, plaques, crusts, and nodules, while typical *Leishmania* ulcers are less common [4,7]. The disease can present in different clinical forms, i.e. localized CL (LCL) which is limited to the skin and usually occurs on the site of the sandfly bite, mucocutaneous leishmaniasis (MCL) which affects the mucosa and can lead to mutilation, and diffuse CL (DCL) with multiple, diffuse and often larger sized lesions [8].

CL treatment depends on the clinical type and complexity of the CL lesions and availability of treatment options. LCL can be treated with cryotherapy intralesional sodium stibogluconate (SSG), topical agents (paromomycin) [9], and local treatment such as thermotherapy [10] which avoid potentially toxic side-effects related to systemic administration of anti-

leishmaniasis treatments [7–10]. In Ethiopia, all four local treatment options are recommended in the national leishmaniasis treatment guideline, but evidence to show the effectiveness of each treatment option is necessary. SSG is the most widely available treatment option, which is made available for free by the government, but good data on treatment outcomes of intralesional (IL) SSG is not available.

Routinely, LCL patients in Ethiopia receive IL SSG once every two weeks, for a maximum of six doses [11]. The recommendation on the usage of IL SSG for LCL lesions caused by *L aethiopica* is based on results from studies performed in old world [12–13] and new world countries in a highly controlled research setting [14–15]. Results from a study in which patients received IL SSG every week for 12 weeks showed that 100% of patients were cured after 6 weeks. Outcomes seem to depend on causative species, as results from Sri Lanka where L. donovani species is present show very different cure rates than country old world, where *L. aethiopica* species is common. Anecdotal evidence from physicians in Ethiopia indicates that outcomes in Ethiopian patients are worse. There are no published reports that show the effectiveness of IL SSG when given by itself in Ethiopian CL patients, although several studies have looked at effectiveness of systemic SSG [7] or combination treatments [13–14]. Many studies have reported adverse events when treating with systemic SSG [16], but to which extent side effects occur when using IL SSG is less clear.

Therefore, evidence on the effectiveness of IL SSG for the treatment of CL caused by *L. aethipica* is needed to reduce the assumption-related provision of IL SSG for LCL. Therefore, we conducted a study embedded in routine care in Northern Ethiopia, Boru Meda Hospital to study the effectiveness of IL SSG for LCL and to determine how many patients reported adverse events.

## Methods

### Ethics statement

Ethical clearance was obtained from Wollo University (Approval number(s) CMH 5030/13/13), college of medicine and health sciences, institutional review board, and Amhara public health institute (APHI). A support letter was obtained from Boru Meda general hospital. Written informed consent was obtained from all participants, and assent was collected from children aged 12–17 and a written consent was obtained from the children's parent's or guardians.

### Study setting

This study was conducted in Boru Meda General Hospital in the East of the Amhara region which serves as a referral treatment center for CL patients.

### Routine diagnosis

In Boru Meda general hospital CL diagnosis is carried out routinely using SSS microscopy [3]. The diagnosis of CL was done using clinical diagnosis complemented with SSS [8], and Fine Needle Aspiration Cytology (FNAC). When the SSS is negative and the clinical diagnosis is not clear, FNAC or biopsy may be done to confirm CL [17].

### Treatment and follow-up

CL treatment in Boru Meda hospital follows the national treatment guidelines [6–8] and expert opinion or decisions by clinicians/dermatologists [3]. Generally, patients with lesions of a size less than or equal to 5cm, less than 4 lesions, lesions that do not affect the mucosa or joints specifically are assigned to local treatment. There are two local treatment options for LCL, IL SSG

is the most widely available one, and cryotherapy is given only when it is available. In our site, giving cryotherapy rather than SSG IL is preferred in patients for whom cryotherapy is possible (lesion not on the nose, size <3 cm, etc.). Unfortunately, cryotherapy was available for only a month duration within the data collection period due to running out of the liquid nitrogen in the hospial.During this period, only 17 patients were treated using cryotherapy.

According to expert opinion and abroad-based research recommendations [15–16], IL SSG is given up to six cycles, administered once in a couple of weeks.In this study the patients had SSG IL once per week for six weeks then after they had outcome assessment at D90. For the outcome variables, only the CL patients who came for their outcome visits on day 35 and day 90 were analyzed. Patients who had IL SSG less than 4 doses or lost at day 35 or 90 were excluded from the outcome analysis. Patients who were not cured at D90 after getting six doses of treatment, were switched to systemic treatment option [15–16].

## Study design, population, and recruitment

This study was conducted using a pragmatic type of study design among patients with LCL who received IL SSG after clinical and laboratory decisions. Patients were asked if they wanted to participate in the study after they had been diagnosed The patient recruitment was started at the date of diagnosis but before administration of IL SSG. Sample size calculation was done based on estimation of the proportion of patients cured (effectiveness) at day 90. Using an estimated cured proportion of 0.70, a desired precision of 0.1 and an 95% confidence level, the required sample size is 81 patients. For this study, we followed the pragmatic trial extension of the Consolidated Standards of Reporting Trials (CONSORT) guidelines (S1 CONSORT Checklist) [17].

## Study procedures

The decision to provide IL SSG was part of routine clinical care. Detailed information was collected at recruitment and patient outcomes were recorded weekly during the 35 days treatment period and at a final assessment 90 days (+/- 1 week) after the start of treatment. Patients who were cured before finishing six weeks of treatment did not receive additional treatments but were asked to come weekly for the follow up visits and for the final outcome visit.

Patients who previously had a positive SSS had a repeated SSS at every consecutive visit until the SSS became negative. Double slide reading was done for quality purposes. Side effects were recorded at every treatment visit. They were rated as mild, moderate, and severe according to Common terminology criteria for adverse events (CTCAE).

## Data collection and analysis

Data were collected using pre-tested structured questionnaires to document the effectiveness of the IL SSG on LCL by 5 trained data collectors (2 dermatologists, 1 trained health officer, and 3 BSc nurses). Data was checked every day and before data entry. The dermatologist diagnosed the patients from the two OPDs. Henceforth, the patients with LCL was assigned randomly after recruitment. Then the patient were took their treatments weekly pattern. But those patients who has lost more than their IL SSG treatment, they were cancelled from the study.

## Data analysis

Data were entered and cleaned using Statistical Package for Social Science (SPSS) IBM Corp. Released 2011. IBM SPSS Statistics for Windows, Version 20.0. Armonk, NY: IBM Corp.

STATS: StataCorp. 2015. Stata Statistical Software: Release 14. College Station, TX: StataCorp LP. Analysis was done using Stata 14. Descriptive statistics were done with medians and inter-quartile ranges and numbers and proportions.

For the outcome data, a per protocol analysis was done including all patients who had received at least 4 doses of SSG IL and who came for their outcome visit. The outcome measures were described using the proportion of cure with the 95% CI. A statistical model was developed to explore determinants of cure of IL SSG treatment for LCL. Definitions of the different outcome categories are described below. To assess factors associated with cure, a multi-level logistic regression model was built using cure/no cure at day 90 as the outcome variable, with patients whose outcome cure classified as no cure.

To get the model that best fits the data, the null model was computed to analyze the relevance of using the mixed model with the Intra-class correlation coefficient (ICC) calculated from it. Later, the null model would also serve as a baseline model for comparison with final multilevel models. Then a model containing the number/size of the lesion as its fixed and random effect was determined before the final model containing the fixed effects of variables of interest, the random intercept was fitted. The model selection strategy used was comparing the Akaike Information Criterion (AIC) and Bayesian information criteria (BIC) of models considered, with the final model having the smallest AIC and BIC value. A significance level of 0.05 was taken as a cut point for all statistical tests.

## Definitions

**Cure.** Defined as 100% epithelization (if ulceration), or 100% flattening (if no ulceration) of the index lesion [14], and if no parasites are seen on the SSS microscopy [18,19].

**Good treatment response.** When one or more of the following criteria are met the patient is considered to have a good response: 1) When the skin slit result decreased in grade [19]. 2) One of multiple lesions has cured. 3) 50–99% reepithelization (if ulceration) or flattening of the index lesion size, with the worst response being binding (if 100% flattening and 60% reepithelization it is considered good response) [14].

**Partial treatment response.** 1–49% reepithelization (if ulceration), flattening of the index lesion [14]. In addition, when there is no decrease report in parasitological from the previous SSS gradings [19].

**No treatment response.** When there is 0% reepithelization (if ulceration), or 0% flattening of the index lesion [14]. In addition, when there is no decrease in SSS grading [19].

**Adherence.** A weekly IL SSG treatment that had at least four doses from the expected six doses, and outcome assessment either at day 35 or 90. When the patients received less than 4 (missing more than 2 prescribed treatments) doses of IL SSG, it will be defined as poor adherence.

**Relapse.** New lesion on the same site or worsening from the previously improving lesion [14,19].

## Results

### Patient characteristics

A total of 83 patients were included in the study, of which 72 completed their treatment according to protocol and came for outcome assessment (Fig 1).

The characteristics of the included patients are described in Table 1. The median age of study participants was 20 years and 41 (56.9%) of them were males. Nearly half (38,4.7%) of the respondents were farmers and 39 (54.2%) of the patients came from the urban area, while the remaining patients came from rural districts.

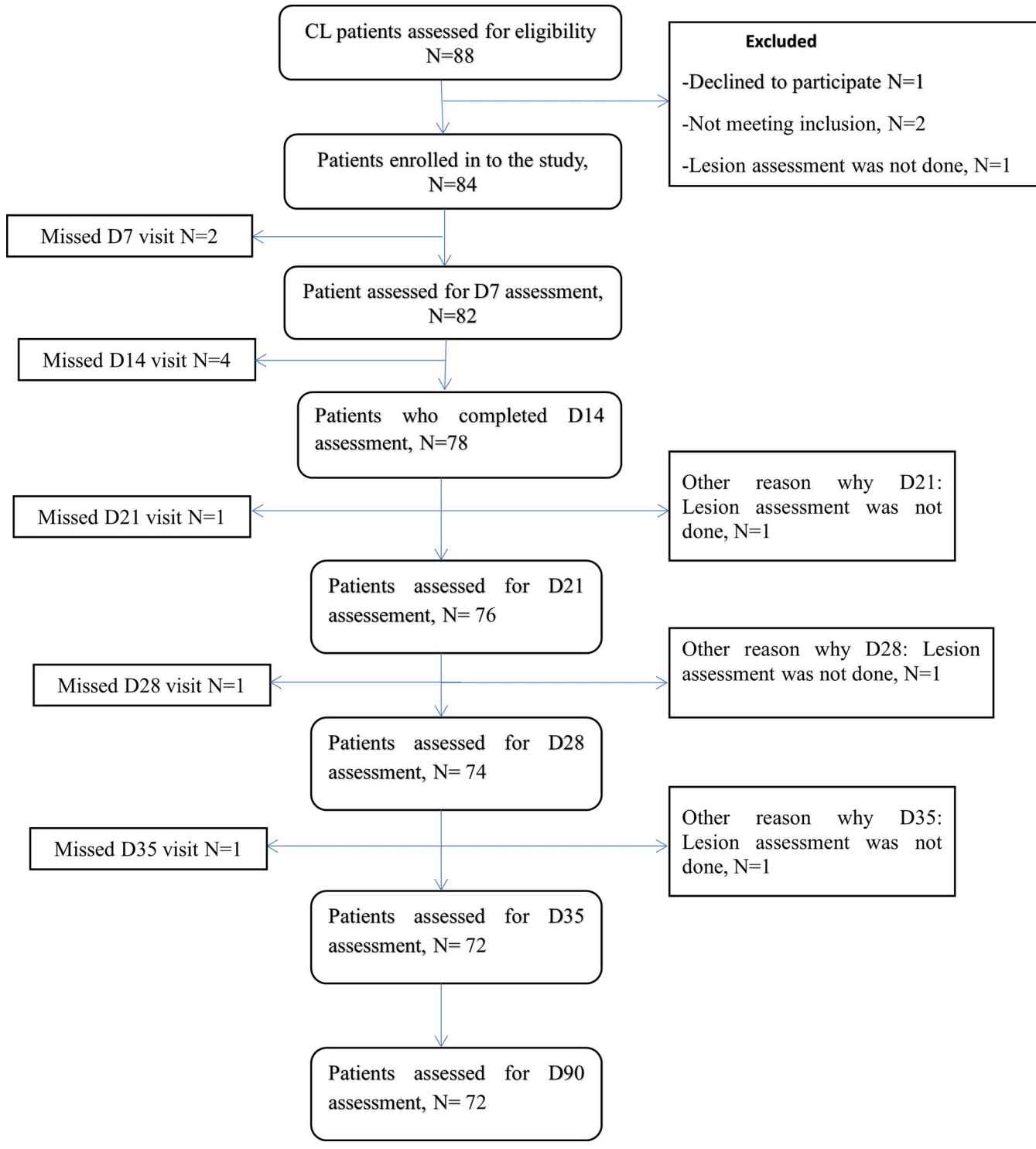

**Fig 1. LCL IL SSG treatment assessement flow chart, Boru Meda General hospital, 2021**

The median size of the lesion was 3cm (IQR 3.2 ± 1.0). Twenty (27.8%) of the patients' index lesion was smaller than 4cm. More than half (40, 55.6%) of the patients had a single lesion and 27 (37.5%) of the patients had an index lesion found on the face.

**Table 1. Patient characteristics of LCL patients on IL SSG treatment, Boru Meda General Hospital, 2021.**

| | | Number (%) |
|---|---|---|
| Variable | Category | N = 83 |
| **Socio-demographic characteristics** | | |
| Age, median (IQR) | 20 (3 ± 2) | |
| Sex | Male | 47 (55.3) |
| | Female | 36 (44.7) |
| Occupation | Farmer | 38 (44.7) |
| | Student | 21 (24.7) |
| | Daily Worker | 9 (10.6) |
| | Civil servant | 7 (8.2) |
| | Others | 8 (11.8) |
| Educational background | Illiterate | 11 (12.9) |
| | Elementary | 35 (41.2) |
| | High school and preparatory | 22 (25.9) |
| | Diploma and above | 15 (20) |
| **Clinical characteristics** | | |
| Duration of the lesion (months) | ≤6 | 9 (10.6) |
| | 7–12 | 23 (27.1) |
| | 13–24 | 42 (49.4) |
| | 25–36 | 9 (12.9) |
| Number of lesions | One | 50 (59.6) |
| | Two | 25 (31.2) |
| | Three | 8 (9.2) |
| Site of the lesion | Face | 29 (34.1) |
| | Earlobe | 8 (9.4) |
| | Extremities | 27 (32.7) |
| | Neck | 6 (7.1) |
| | More than 2 sites | 13 (16.7) |
| Traditional treatment | Yes | 61 (73.5) |
| | No | 22 (26.5) |

A total of 57 (79.2%) of LCL patients had already used both traditional and modern treatments. Three-quarters of the patients had a history of prior use of traditional agents from the traditional healers. More than half (44, 61.1%) of the LCL patients came after already using treatments from the referring health institution. Most of these (38, 52.8%) used topical antifungals and steroids (were therefore mistreated).

At recruitment, 61 (84.7%) patients had a sample that was positive for SSS; 10 (13.9%) had a grade of +1, 10 (13.9%) had a grade of +2, and the rest (35.7%) were +3 SSS grading. A negative skin slit smear was found in 11 (15.3%) patients, among these 4 were positive using FNAC and the remaining 7 were therefore treated empirically. When we see the treatment response among the 7 unconfirmed patients, the cure rate was not significantly different (63.5% for confirmed and 55.9% for unconfirmed patients).

## Treatment and outcomes

A total of 11 patients received less than 4 doses of treatment (without reaching cure) and was considered as lost-to-follow-up. The remaining 72 patients (86.7) were considered as having good adherence with 24 patients receiving 5 doses and 48 patients completing their 6 doses of

**Table 2. Skin slit smear test result grading for localized cutaneous leishmaniasis patients, Boru Meda general Hospital, 2021.**

| Skin slit smear Grading | D0 (n, %) | D35 (n, %) | D90 (n, %) |
|---|---|---|---|
| +1 | 10 (13.9) | 21 (29.2) | 18 (25) |
| +2 | 10 (13.9) | 2 (2.8) | 3 (4.2) |
| +3 | 25 (34.7) | 4 (5.6) | 4 (5.6) |
| +4 | 9 (12.5) | 0 | 0 |
| +5 | 4 (5.6) | 0 | 0 |
| +6 | 3 (4.2%) | 0 | 0 |
| Total positive, n (%) | 61 (84.7) | 27 (37.5) | 25 (34.7) |

treatment. The median number of SSG IL treatments given to patients was 5 (IQR 5–6). Side effects were common with 95.8% of the patients experiencing injection site pain with 59 (81.9%) having mild, 10 (13.9%) moderate, and 3 (4.2%) severe pain with systemic manifestations like fever, injection site edema, vomiting, and pain to touch.

After six injections at day 35, 27 (37.5%) of patients were still positive on the SSS, with grades ranging from +1 (21, 29.2%) to +3 (4, 5.6%). At day 90, 25 (34.7%) patients were still positive for the SSS (Table 2).

Amongst 83 included patients, 82 (98.8%), 78 (94%), 76 (91.6%), 74 (89.2%), and 72 (86.8%) patients came for their day 7, 14, 21, 28, and 35 assessment respectively (Fig 1). 72 completed treatment and came for outcome assessment at day 90. Of those 72, only 43 (59.7%, 95% CI 0.44–0.69) cases were cured, 10 (13.9%) had good response, 12 (16.9%) had a partial response, 2 (2.8%) of cases showed no response and 5 (6.9%) worsened (Table 3). Patients who were not cured were started on systemic treatment 13/29 (44.8%) and 16/29 (55.2%) were provided a follow-up appointment for re-evaluation.

Multilevel logistic regression was done to identify factors associated with cure at day 90. In the final model, age, size of the lesion, sex, SSS grading before treatment, and educational status were significantly related to treatment outcome. Increasing age (OR 1.07, 95% CI: 1.02–1.27), being male (OR 1.79, 95% CI: 1.07–2.25), having SSS grading of +1, (OR 1.53, 95% CI 1.24–1.73) and +2, (OR 1.51, 95% CI: 1.41–3.89) had higher chance of cure, while increasing lesion size decreased chance of cure (OR 0.79, 95% CI: 0.08–0.94). The number of lesions, SSS positive grading, and morphology of the lesion were not found to be independent predictors of the effectiveness of the IL SSG for LCL (Table 4).

## Discussion

The overall cure rate for treatment of LCL with IL SSG was around 60% whereas 10 (13.9%) had a good response, 12 (16.9%) partial response, 2 (2.8%) no response, and 5 (6.9%)

**Table 3. IL SSG treatment outcomes among LCL patients on IL SSG treatment, Boru Meda General Hospital, 2021.**

| Outcome | D7 N = 82 | | D14 N = 78 | | D21 N = 76 | | D28 N = 74 | | D35 N = 72 | | D90 N = 72 | |
|---|---|---|---|---|---|---|---|---|---|---|---|---|
| | N (%) | CI (95%) | N (%) | CI (95%) | N (%) | CI (95%) | N (%) | CI (95%) | N (%) | CI (95%) | N (%) | CI (95%) |
| Worsening | 2 (2.8) | 0.0–6.9 | - | - | 1 (1.4) | 0.0–4.4 | 4 (5.6) | 0.0–11.1 | 5 (5.6) | 1.2–14.3 | 5 (6.9) | 1.2–14.3 |
| No responsee | 9 (12.5) | 4.0–20.2 | 15 (20.8) | 13.5–28 | 5 (5.6) | 1.2–12.5 | 2 (2.8) | 0.0–7.1 | 2 (2.8) | 0.00–9.9 | 2 (2.8) | 0.0–7.1 |
| Partial responsee | 49 (68.1) | 56.9–79.4 | 55 (76.4) | 67.9–84.7 | 41 (56.9) | 41.5–67.2 | 35 (48.6) | 36.9–59.5 | 25 (34.7) | 19.4–44.6 | 12 (16.9) | 9.3–27.0 |
| Good responsee | 12 (16.7) | 7.9–23.6 | 2 (2.8) | 0.00–7.1 | 25 (34.7) | 25.0–47.4 | 30 (41.7) | 31.4–54.4 | 37 (51.4) | 37.3–62.9 | 10 (13.9) | 6.2–21.0 |
| Cure | - | - | - | - | - | - | 1 (1.4) | 0.0–5.6 | 5 (5.6) | 1.4–12.5 | 43 (59.7) | 0.44–0.69 |

**Table 4. Multi-level logistic regression or fixed effects on the effectiveness of IL SSG for the treatment of LCL, Boru Meda general hospital, 2021.**

| Variables | | | OR | 95% Conf. Interval | | P value |
|---|---|---|---|---|---|---|
| | | | | Lower | Upper | |
| **Model one (Intial or Null model)** | | | | | | |
| Random-effects Parameters | | | | | | |
| Residual (sd (_cons)) | | | 114.299 | - | - | 0.000 |
| Variance for intercepts | | | 82.396 | 27.825 | 469.523 | 0.000 |
| **Model two** | | | | | | |
| Random-effects Parameters | | | | | | |
| Residual (sd (_cons)) | | | 0.00013 | - | - | 0.000 |
| Variance for intercepts | | | 1.58 | 5.73e-09 | 5.73e-09 | 0.000 |
| Size of the lesion | | | 1.28 | 1.078 | 3.718 | 0.071 |
| Number of the lesion | | 1. One | 2.704 | 1.662 | 9.610 | 0.681 |
| | | 2. Two | 1.800 | 2.104 | 11.572 | 0.491 |
| Age | | | 1.073 | 1.022 | 1.125 | 0.004 |
| 1. Sex | | | 7.83 | 1.542 | 39.771 | 0.013 |
| Occupation | 1. Civil servant | | 2.18e+08 | - | - | 0.989 |
| | 2. Student | | 8.759 | 1.047 | 73.266 | 0.504 |
| | 3. Daily worker | | 1.306 | 0.136 | 12.57 | 0.817 |
| | 4. Merchant | | 8.59e-08 | - | - | 0.994 |
| | 5. Other | | 0.124 | 0 .003 | 4.233 | 0.246 |
| Educational level | 1.Diploma and above | | 1.019 | 1.0006 | 1.569 | 0.022 |
| | 2.Elementary | | 1.41 | 0.207 | 9.589 | 0.725 |
| | 3.High school & preparat | | 0.547 | 0.0625 | 4.783 | 0.585 |
| Test result positive grading | 1 | 1.88 | | 3.84 | 11.91 | 1.350 |
| | 2 | 2.2 | | 2.17 | 8.57 | 0.127 |
| | 3 | 6.1 | | 5.06 | 7.036 | 0.461 |
| | 4 | 0.7 | | 0.06 | 1.33 | 0.710 |
| _cons | | 0.0686 | | 0.006 | 0.833 | 0.385 |
| **Model three (combined)** | | | | | | |
| Random-effects Parameters | | | | | | |
| Residual (sd (_cons)) | | | 2.441 | - | - | |
| Variance for intercepts | | | 3.180 | 0 .189 | 31.37 | 0.29 |
| Number of the lesion | | 1. One | 1.107 | 0.027 | 45.668 | 0.957 |
| | | 2. Two | 1.545 | 0.029 | 80.469 | 0.829 |
| Size of the lesion | | | 0.79 | 0.078 | 0.936 | 0.003* |
| Educational level | 1.Diploma and above | | 0.091 | 0.0003 | 24.78 | 0.403 |
| | 2.Elementary | | 3.118 | 0.074 | 131.268 | 0.551 |
| | 3.High school & preparatory | | 0.483 | 0.014 | 17.017 | 0.689 |
| Age of the patient | | | 1.069 | 1.018 | 1.266 | 0.004* |
| Sex | | | 1.32 | 1.069 | 2.25 | 0.004* |
| Occupation | Civil servant | | 9.38 | 5.239 | 6.401 | 1.028 |
| | Student | | 3.851 | 1.403 | 3.528 | 0.774 |
| | Daily worker | | 1.526 | 1.136 | 2.061 | 0.661 |
| | Merchant | | 11.11 | 9.125 | 11.204 | 0.814 |
| | Other | | 1.290 | 2 .107 | 8.041 | 0.920 |
| Test result positive grading | 1 | | 1.53 | 1.24 | 1.731 | 0.040* |
| | 2 | | 1.41 | 1.51 | 3.89 | 0.005* |
| | 3 | | 0.698 | -0.927 | 0.207 | 0.210 |
| | 4 | | 0.717 | -0.952 | 0.286 | 0.286 |

(*Continued*)

**Table 4.** (Continued)

| Variables | OR | 95% Conf. Interval | | P value |
|---|---|---|---|---|
| | | Lower | Upper | |
| _cons | 0.0002 | 9.81e-12 | 2976.363 | 0.308 |

Associated variables

worsened. Injection site pain was reported as a side effect by almost all patients, but this was mostly only mild. Male sex, age, size of the lesions, and SSS result grade were significantly related to cure at day 90, with males, older patients, and patients with smaller lesions having a higher chance of cure.

The cure rate of IL SSG in the study area was 59.7% which is relatively low compared to studies done in most other countries. Only in a study conducted in Isfahan, Iran a lower cure rate of 41.7% was found [1], although this study had very strict criteria for a cure by requiring lesion to be completely healed without scar in less than two months. Our results were consistent with a study conducted in Bolivia where patients received either three SSG injections in five days with a cure rate of 57% at six months or five injections in 11 days with a cure rate of 73% [11]. Since this study was a clinical trial, it has more strict inclusion criteria such as a maximum of one lesion of a maximum of 3cm and therefore the study population was probably less severe. In contrast, our results are much lower than those from Sri Lanka, where a 96.3% cure rate was observed, although they also graded 80–90% improvement as cure [12]. In Iraq, 97.2% of patients were cured six months after receiving a total of ten IL SSG injections in five weeks [20].

Higher cure rates in other studies may be caused by a less severe patient population, as almost half presented within the first month of symptoms, and only patients with ≤3 lesions of ≤3 cm diameter were treated with IL SSG. A combination of factors is likely responsible for our relatively low cure rate, including outcome assessment at day 90 instead of day 180, a patient population with longstanding lesions, and relatively strict criteria for cure. However, this is the typical CL patient population in Ethiopia [7], and therefore we stress that it is important to do effectiveness studies on real-life patient populations in addition to controlled clinical trials. The causative species is also likely to play a role, as it seems that *L. aethiopica* is relatively hard to treat compared to other species [21].

The treatment schedule of IL SSG administration for CL may also be important, as a wide variety of schedules is used throughout the world, ranging from daily, to every two or three weeks [16–17]. Not many studies have been done to compare treatment schedules, but twice weekly treatment showed significantly higher and earlier cure rates in India than weekly. Moreover, the provision of IL SSG in alternate-day and weekly schedules also had higher cure rates than daily injections [16]. Perhaps twice-weekly injections of IL SSG could improve the cure rates that were observed in this patient population, but having to come twice per week for treatment may prove to be practically difficult for patients in a country like Ethiopia.

Data from the SSS showed that even after multiple sessions of treatments, patients stay microscopy positive. Among 61 (84.7%) positive at D0, 25 (34.7%) patients were still positive. But, 7 patients became positive at D90 even though they were negative at D0 and 4 cases were clinically cured but still positive. Several other studies have shown parasite persistence long after treatment [22–23]. In general, it is recommended to use clinical outcomes to assess cure [19], but in practice, skin slit is often still done to monitor outcomes and to decide whether to extend treatment in Ethiopia [24]. Two studies using molecular techniques have shown that measurable parasite load weeks after treatment was related with treatment outcomes [20,23],

and therefore it would be worthwhile to further explore the clinical value of a persistingly positive SSS after treatment.

The results from this study showed that increasing age and being male were associated with a higher chance of cure after IL SSG treatment of LCL caused by *L. aethiopica*. Increasing age being related to better cure rates was also observed in several other studies [15], of which one investigated systemic miltefosine treatment in Ethiopia [19], and two studies were done in Latin America and studied antimony treatment [20]. The association between increasing lesion size and lower cure rates was also seen in the Ethiopian paper, while a study from Brazil similarly showed that lesions <1 cm in size were associated with cure [25,26]. No other studies found [27] male sex to be related to cure rate, on the contrary, one study from Brazil showed higher levels of cure in females patients [25,28].

In this study, we systematically recorded treatment outcomes of IL SSG for the first time in Ethiopia in a real-life population. Despite a relatively low sample size, we had good data from patient follow-up visits. Ideally, we would have had outcome data from a longer follow-up period up to six months, but this was not possible due to practical reasons.

## Conclusion

Our findings showed that 6 weekly doses of the treatment IL SSG resulted in a cure rate of around 60% and that injection site pain was experienced by almost all patients. Therefore, other treatment options should be explored and compared in clinical trials to identify which treatment for LCL should be recommended in Ethiopia.

## Supporting information

**S1 CONSORT Checklist. CONSORT Statement—Checklist of items for reporting pragmatic trials**
(DOCX)

**S1 Tables Table A. Measurements of LCL Before and while ongoing treatment (follow ups), Boru meda general Hospital, 2021.** Table B: Lesion summary tables of missed follow-up LCL patients, Boru Meda general hospital, 2021. Table C: Missed follow-up patients' description and clinical profiles for IL SSG treating LCL, Boru Meda general hospital, 2021. Table D: The fit and residual values of the models of effectiveness of IL SSG treating LCL, Boru meda general hospital, 2021.
(DOCX)

## Acknowledgments

My appreciation goes to study participants and I would like to thank the data collectors (dermatovenereologists, health officers, and nurses in the dermatology clinic) for their great efforts.

## Author Contributions

**Conceptualization:** Feleke Tilahun Zewdu.

**Data curation:** Feleke Tilahun Zewdu, Asressie Molla Tessema.

**Formal analysis:** Feleke Tilahun Zewdu, Asressie Molla Tessema.

**Investigation:** Feleke Tilahun Zewdu, Aregash Abebayehu Zerga.

**Methodology:** Feleke Tilahun Zewdu, Asressie Molla Tessema.

**Software:** Feleke Tilahun Zewdu, Asressie Molla Tessema, Aregash Abebayehu Zerga.

**Supervision:** Aregash Abebayehu Zerga.

**Validation:** Asressie Molla Tessema, Aregash Abebayehu Zerga, Saskia van Henten.

**Visualization:** Saskia van Henten, Saba Maria Lambert.

**Writing – review & editing:** Feleke Tilahun Zewdu, Saskia van Henten, Saba Maria Lambert.

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
