## [Decision Letter · Decision Letter 0]

21 Feb 2022

Dear Mr Tilahun,

Thank you very much for submitting your manuscript "Effectiveness of intralesional sodium stibogluconate (IL SSG) for the treatment of localized cutaneous leishmaniasis at Boru Meda general hospital, Amhara, Ethiopia: Pragmatic trial" for consideration at PLOS Neglected Tropical Diseases. As with all papers reviewed by the journal, your manuscript was reviewed by members of the editorial board and by several independent reviewers. In light of the reviews (below this email), we would like to invite the resubmission of a significantly-revised version that takes into account the reviewers' comments. 

We cannot make any decision about publication until we have seen the revised manuscript and your response to the reviewers' comments. Your revised manuscript is also likely to be sent to reviewers for further evaluation.

Sincerely,

Christine A Petersen

Deputy Editor

Christine Petersen

Deputy Editor

Reviewer's Responses to Questions

**Key Review Criteria Required for Acceptance?**

**Methods**

-Are the objectives of the study clearly articulated with a clear testable hypothesis stated?

-Is the study design appropriate to address the stated objectives?

-Is the population clearly described and appropriate for the hypothesis being tested?

-Is the sample size sufficient to ensure adequate power to address the hypothesis being tested?

-Were correct statistical analysis used to support conclusions?

-Are there concerns about ethical or regulatory requirements being met?

Reviewer #1: Please see attached document for comments to the authors.

Reviewer #2: The objectives are well defined, and in general the design used seems correct, using the definition of a pragmatic trial. However the sample size seems small, and should be calculated or explained. Statistical analysis seems correct. 

In page 6, I think that there is a mistake about the IL SSB schedule, that was administered weekly and not every 2 weeks. Please correct.

I have no concerns about ethical in this study.

Reviewer #3: Objective of this study is very nicely explained. Study Design appropriate, however I have few minor comments.

1. Please provide the details of the procedure for administration of intralesional SSG (Line 107-117).

2. Please provide the limitation or success of other treatment approaches (Line 70-74). Authors has mentioned that thermotherapy/ cryotherapy or paramomycine treatment is recommended, however it is not clear if these study not done yet or these treatments are not effective; please make this statement clear.

**Results**

-Does the analysis presented match the analysis plan?

-Are the results clearly and completely presented?

-Are the figures (Tables, Images) of sufficient quality for clarity?

Reviewer #1: Please see attached document for comments to the authors.

Reviewer #2: Results are well described and analised. However, in page 8, authors should give more details about "traditional and modern treatments". In page 10, it should be informed dosage and schedule of systemic treatment for those who were not cured. Table 1 should provide information regarding what the authors call "traditional tretament".

Reviewer #3: Results were very clearly presented

**Conclusions**

-Are the conclusions supported by the data presented?

-Are the limitations of analysis clearly described?

-Do the authors discuss how these data can be helpful to advance our understanding of the topic under study?

-Is public health relevance addressed?

Reviewer #1: Please see attached document for comments to the authors.

Reviewer #2: In discussion, page 14, lines 311-313 the sentence is confusing. Please explain better.

Conclusion is supported by the data and public health relevance is addressed.

Reviewer #3: Conclusion supported by the data presented. This study definitely addressed solution for the LCL caused by Leishmaniasis infection which is a real public health challenge at Ethiopia.

**Editorial and Data Presentation Modifications?**

Reviewer #1: Please see attached document for comments to the authors.

Reviewer #2: In general the modifications required are few, so I think that minor revision applies. Authors should also inform in the abstract and in page 4 (first paragraph) the cure rate of systemic treatment or what they mean with "relatively hard to treat" (abstract) or "difficult to treat" (page 4).

Reviewer #3: This study is very important addition for the Leishmania field. I would recommend for accepting this article after minor revision.

**Summary and General Comments**

Reviewer #1: Please see attached document for comments to the authors.

Reviewer #2: It is an important contribuiton showing that IL SSB unfortunately has a low cure rate and important morbidity in CL at Ethiopia, and therefore patients deserve a better therapy. The pragmatic design is well justified in this setting. All modifications required are listed in other sections.

Reviewer #3: (No Response)

PLOS authors have the option to publish the peer review history of their article (what does this mean?). If published, this will include your full peer review and any attached files.

Reviewer #1: No

Reviewer #2: No

Reviewer #3: Yes: Ranadhir Dey
---

## [Decision Letter · Decision Letter 1]

11 Jun 2022

Dear Mr Tilahun,

We are pleased to inform you that your manuscript 'Effectiveness of intralesional sodium stibogluconate for the treatment of localized cutaneous leishmaniasis at Boru Meda general hospital, Amhara, Ethiopia: Pragmatic trial' has been provisionally accepted for publication in PLOS Neglected Tropical Diseases.

Best regards,

Christine A. Petersen

Deputy Editor

Christine Petersen

Deputy Editor

Reviewer's Responses to Questions

**Key Review Criteria Required for Acceptance?**

**Methods**

-Are the objectives of the study clearly articulated with a clear testable hypothesis stated?

-Is the study design appropriate to address the stated objectives?

-Is the population clearly described and appropriate for the hypothesis being tested?

-Is the sample size sufficient to ensure adequate power to address the hypothesis being tested?

-Were correct statistical analysis used to support conclusions?

-Are there concerns about ethical or regulatory requirements being met?

Reviewer #1: All reviewer comments were addressed appropriately.

Reviewer #2: No comments.

**Results**

-Does the analysis presented match the analysis plan?

-Are the results clearly and completely presented?

-Are the figures (Tables, Images) of sufficient quality for clarity?

Reviewer #1: All reviewer comments were addressed appropriately.

Reviewer #2: No comments.

**Conclusions**

-Are the conclusions supported by the data presented?

-Are the limitations of analysis clearly described?

-Do the authors discuss how these data can be helpful to advance our understanding of the topic under study?

-Is public health relevance addressed?

Reviewer #1: All reviewer comments were addressed appropriately.

Reviewer #2: No comments.

**Editorial and Data Presentation Modifications?**

Reviewer #1: All reviewer comments were addressed appropriately.

Reviewer #2: No comments.

**Summary and General Comments**

Reviewer #1: All reviewer comments were addressed appropriately.

Reviewer #2: No comments.

PLOS authors have the option to publish the peer review history of their article (what does this mean?). If published, this will include your full peer review and any attached files.

Reviewer #1: No

Reviewer #2: No

---

## [Editor Report · Acceptance letter]

27 Aug 2022

Dear Mr Tilahun,

We are delighted to inform you that your manuscript, "Effectiveness of intralesional sodium stibogluconate for the treatment of localized cutaneous leishmaniasis at Boru Meda general hospital, Amhara, Ethiopia: Pragmatic trial," has been formally accepted for publication in PLOS Neglected Tropical Diseases.

Best regards,

Shaden Kamhawi

co-Editor-in-Chief

Paul Brindley

co-Editor-in-Chief
